# Hippocampal representations emerge when training recurrent neural networks on a memory dependent maze navigation task

## Abstract

Can neural networks learn goal-directed behaviour using similar strategies to the brain, by combining the relationships between the current state of the organism and the consequences of future actions? Recent work has shown that recurrent neural networks trained on goal based tasks can develop representations resembling those found in the brain, entorhinal cortex grid cells, for instance. Here we explore the evolution of the dynamics of their internal representations and compare this with experimental data. We observe that once a recurrent network is trained to learn the structure of its environment solely based on sensory prediction, an attractor based landscape forms in the network's representation, which parallels hippocampal place cells in structure and function. Next, we extend the predictive objective to include Q-learning for a reward task, where rewarding actions are dependent on delayed cue modulation. Mirroring experimental findings in hippocampus recordings in rodents performing the same task, this training paradigm causes nonlocal neural activity to sweep forward in space at decision points, anticipating the future path to a rewarded location. Moreover, prevalent choice and cue-selective neurons form in this network, again recapitulating experimental findings. Together, these results indicate that combining predictive, unsupervised learning of the structure of an environment with reinforcement learning can help understand the formation of hippocampus-like representations containing both spatial and task-relevant information.

## 1 Introduction

Recurrent neural networks have been used to perform spatial navigation tasks and the subsequent study of their internal representations has yielded dynamics and structures that are strikingly biological. Metric (Cueva & Wei, 2018; Banino et al., 2018) and non-metric (Recanatesi et al., 2019) representations mimicking grid (Fyhn et al., 2004) and place cells (O'Keefe & Nadel, 1978) respectively form once the recurrent network has learned a predictive task in the context of a complex environment. Cueva et al. (2020) demonstrates not only the emergence of characteristic neural representations, but also hallmarks of head direction system cells when training a recurrent network on a simple angular velocity integration task. Biologically, non-metric representations are associated with landmark spatial memory, in which place cells within the mammalian hippocampus fire when the associated organism is present in a corresponding place field. Extrafield firing of place cells occurs when these neurons spike outside of these contiguous place field regions. Here we show that recurrent neural networks (RNNs) not only form corresponding attractor landscapes, but also produce representations with internal dynamics that closely resemble those found experimentally in the hippocampus when performing goal-directed behaviour.

Research in neuroscience such as that of Johnson & Redish (2007), shows that spatial representations in mice in the CA3 region of the hippocampus frequently fire nonlocally. Griffin et al. (2007) show that a far higher proportion of hippocampal neurons in the CA1 region in rats performing an episodic task in a T-shaped maze encode the phase of the task rather than spatial information (in this case trajectory direction). Ainge et al. (2007) show CA1 place cells encode destination location at the start position of a maze. Lee et al. (2006) demonstrate that place fields of CA1 neurons gradually drift toward reward locations throughout reward training on a T-shaped maze.

In this work we show that a recurrent neural network learning a choice-reward based task using reinforcement learning, in conjunction with predictive sensory learning in a T-shaped maze produces an internal representation with consistent extrafield firing associated with consequential decision points. In addition we find that the network's representation, once trained, follows a forward sweeping pattern as identified by Johnson & Redish (2007). We then show that a higher proportion of units in the trained network show strong selectivity for the encoding or choice phase of the task than the proportion showing selectivity for spatial topology. Importantly, these properties only emerge during predictive learning, where task learning is much faster compared to traditional deep Q learning.

## 2 METHOD

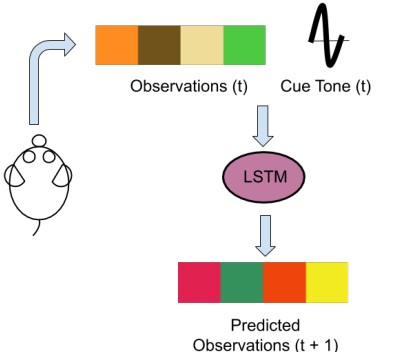 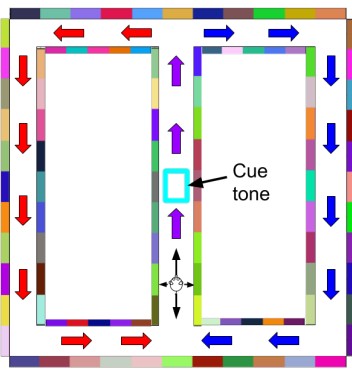

Figure 1: Left, the wall observation and cue received by the network at each timestep. Right, the entangled predictive task the LSTM network is pre-trained on in order to generate a non-metric map of the maze environment.

We use a form of the cued-choice maze used by Johnson & Redish (2007) which has a central T structure with returning arms, shown in Figure 1. All walls of the maze are tiled with distinct RGB colours which are generated at random and remain fixed throughout. An agent is initially learning to predict the next sensory stimulus given its movement. This combination of unsupervised learning and exploration has been shown previously to produce place cell-like encoding of the agent's position (Recanatesi et al., 2019). Next, rewards at four possible locations are introduced and the agent is tasked with associating a cue with the rewarding trajectory. The agent has four vision sensors, one in each cardinal direction, reading the wall RGB colours they intersect. The cue tone is played to the agent as it passes the halfway point of the central maze stem. A low frequency cue indicates that the agent will turn left at the top of the maze stem and a high frequency cue indicates a right turn. These cue tones take the form of a high or low valued scalar perturbed with normally distributed noise if at a cue point, with a zero value given at all other locations. These four RGB colours as well as the cue frequency at the current location make up the total input received by the agent.

The agent is controlled by a recurrent neural network comprised of a 380 unit Long-Short term memory (Hochreiter & Schmidhuber, 1997) (LSTM) network with a single layered readout for the prediction of RGB values. We first pre-train the network by tasking it with predicting the subsequent observation of wall colours from the currently observable wall colours given its trajectory through the maze. The agent's starting location is at the bottom of the central stem of the T maze and a trajectory of left or right at the top of the central stem is chosen pseudorandomly, depicted with red and blue arrows respectively in Figure 1 and corresponding to the low (red trajectory) or high (blue trajectory) cue tone value given halfway up the stem. As in the experiments by Johnson & Redish (2007), during pre-training the agent does not choose any of its actions and is only learning to predict the sequence of wall colors it encounters. In a given pre-training iteration, we collect all observations as the agent traverses the maze until it returns to the start location at the bottom of the central stem and finally train the LSTM on the entire collected trajectory. The network is trained with a mean-squared error loss of predicted and target wall colours (Eq. 1), with model parameters optimised using Adam (Kingma & Ba, 2015) and a learning rate of 0.001.

$$loss_{rgb} = \frac{1}{n} \sum_{i=1}^{n} (y_{rgb} - (W_{rgb}h_t + b_{rgb}))^2 \tag{1}$$

To solve this task, the network has to maintain the cue tone played in its internal memory for several time steps in order to predict subsequent wall colours from the top of the central stem. In our model, this is achieved through the network forming a non-metric representation (attractor landscape) of the maze environment, as also demonstrated by Xu & Barak (2020). Similarly, behavioural experiments typically have a comparable familiarisation phase with the environment before reward-based tasks are introduced (Johnson & Redish, 2007; Griffin et al., 2007).

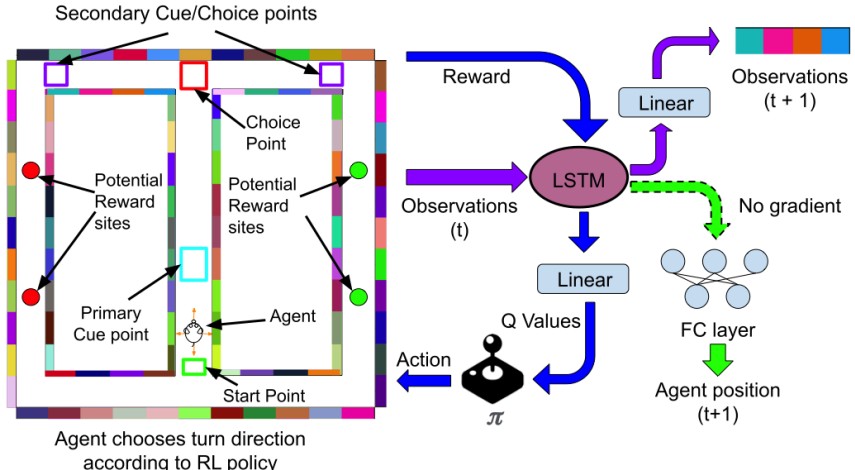

Figure 2: For the joint task to be learned by the LSTM network, we introduce secondary cue points, where the same cue tone as that played at the primary cue point will be repeated if and only if the agent has proceeded in turning in the direction corresponding to the cue tone frequency given at the primary cue location. The agent is free to choose the next action to be taken when traversing the maze at either the choice point at the top of the stem of the maze or at the secondary cue locations. There are two potential reward sites on both returning arms, with the reward sites being active if the agent is on the returning arm corresponding to the cue tone frequency.

Once the LSTM has formed an internal representation of the maze, the agent is tasked with navigating towards potential reward sites whose location is indicated by the cue signal: a low frequency cue indicates active reward sites on the left return arm and a high frequency cue indicates active reward sites on the right return arm - the cue tone and corresponding side of active reward sites are together chosen randomly at each iteration with a secondary cue given if the agent has turned correctly. In this phase there are three choice points wherein the agent is able to choose its next action and is constrained to follow the forward maze direction elsewhere: at the top of the maze stem and at the two secondary choice points (Figure 2), with initially random movement at these points during reward training. There are 5 steps between the cue and choice points and 7 steps from the choice point to the first reward site on either return arm. The inclusion of the secondary cues as additional choice points was motivated by the experimental set up used by Johnson & Redish (2007), to compare the network activity at these points to experimental data. These secondary points also give the agent the opportunity to backtrack on its decision made at the primary choice point in light of further environmental observation (the presentation or lack thereof of the secondary cue), and make learning more efficient in our model. This may explain how it speeds up training the animals in the same task.

We additionally introduce a new single layered readout for the LSTM network which predicts state-action values associated with the four cardinal directions in relation to the agent's current position and direction. At each timestep, this ensemble receives the agent's environment observation and the agent follows an epsilon-greedy policy (starting with fully random movement at choice points and a decaying epsilon thereafter) for choosing optimal actions of those available at each of the three choice points. The recurrent network controlling the agent is trained on a weighted combined loss of a reinforcement learning (RL) task loss and the previously described predictive wall colour loss:

$$loss_{combined} = |Q(s, a) - (r + \gamma \cdot Q'(s', \arg\max_{a'} Q(s', a')))| + \lambda \cdot loss_{rgb} \qquad (2)$$

The first component of this loss is the difference between predicted and observed state-action values which are represented by Q-values (Watkins & Dayan, 1992), which are a prediction of future global reward:

$$Q(s, a) = W_Q h_t + b_Q \qquad (3)$$

We use double-Q learning (Van Hasselt et al., 2016) to train the agent on the task, updating the target Q value predictor ($Q'$ - a LSTM with same number of units) every 15 training iterations. Double-Q learning allows for optimal performance on the reward task in drastically fewer agent maze traversals and network training iterations than with standard DQN (Mnih et al., 2013) based Q-learning which suffers from overestimation of Q-values. We settle on a discount factor ($\gamma$) of 0.8 as values higher than this regularly cause the network to converge on solutions wherein the agent does not take the most direct path to reward locations, with backtracking at secondary choice points. The second loss component is the sensory prediction task which we used to pre-train the network ($\lambda = 0.5$). After the network has been pre-trained (optimising to minimise Eq. 1), the network achieves perfect performance on the predictive task. This loss component is included when training the network on the reward task so that the spatial map (non-metric attractor landscape) of the maze environment formed during pre-training is maintained throughout Q-learning. This ensures the map is not overwritten as would happen when Q-learning is performed alone, and leads to faster task learning (see results). We optimise the network for this joint task using Adam and a learning rate of 0.0005, which we find improves the rate of convergence with optimal task performance, as opposed to higher learning rates which still converge but with backtracking often inherent in task solutions.

In contrast to much of the previous work on spatial representations in recurrent networks, we do not give the network any indication of the agent's location or movement. This makes the task considerably more difficult due to the unpredictable movement possible at choice points during the reward task. The network is coerced into storing the current movement direction of the agent in its representation, in addition to storing the cue frequency. As such, a network of Gated Recurrent Units (Cho et al., 2014) (GRUs) or vanilla RNN units was unable to perform well in either the pre-training or joint RL task due to these prevalent long term dependencies (18 steps between cue and final reward).

To analyse the representations formed by the network, we train a further single layered fully connected network with a softmax layer (shown in green in Figure 2) to predict the agent's next location using the activity of the LSTM. There is no backpropagation of gradients between this predictor and the LSTM network, and the predictor is trained at the end of reward training. The plots in Figures 5 and 6 are distributions indicating the probability of agent location inferred from LSTM activity by the predictor. This is used in place of the decoding algorithm used by Johnson & Redish (2007) to predict the neuronally inferred maze location of rats when performing a cue based task.

## 3 RESULTS

The agent learns the sensory prediction task to a high degree of recall and after around a thousand training iterations (combined loss with pre-training in Figure 3), the agent was able to achieve perfect performance on the reward task when the LSTM network had 380 or more units (Fig. 3, right). We trained the reinforcement learning (Eq. 2) portion of the task in an epsilon greedy manner, with a steadily decaying epsilon to ensure that the agent would choose the rewarding path consistently once actions were chosen at choice points completely by the network. Notably, the agent did not turn at either of the secondary choice points once training had completed - only at the primary choice point.

We attempted to run the reinforcement learning task alone in a maze with no sensory input except the reward cue. In this scenario the network is not able to learn the task due to a lack of self-localisation and is unable to perform the task based on step counting between the cue and choice point. In addition, the reward based reinforcement learning task was attempted using Q-learning alone with a loss function that did not include the wall colour prediction error, both with and without pre-training (shown in Fig. 3, left). In both cases we find that the reward task is not learnable with the same higher rate of epsilon decay we use for the combined loss function with pre-training, as the network quickly forgets the attractor landscape of the maze formed during pre-training, which we maintain through the combined loss (Eq. 2). We also find the network can solve the reward task using the combined loss without pre-training, albeit in around 3 times the number of maze traversals as with the use of the spatial map formed in the pre-trained case (Fig. 3, left).

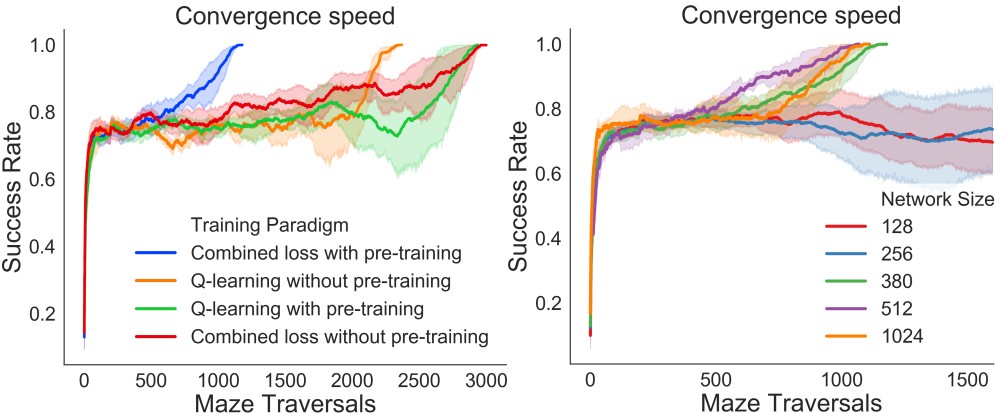

Figure 3: Left: Success rate (proportion of direct traversals to reward locations) of each set of training paradigms on the reward task, averaged over 10 initial conditions and random wall colours using optimal rate of epsilon decay for each paradigm, each shown with a 95% confidence interval. Attractor landscape formed during pre-training alongside combined loss allows network to achieve perfect performance on reward task in relatively few maze traversals. Q-learning alone without pre-training also achieves perfect performance in more than twice the number of maze traversals. Q-learning alone with pre-training takes far more maze traversals to converge (and is less likely to be optimal) due to the non-random initial state of network and inability to utilise the spatial map formed. Combined training without pre-training also takes relatively many maze traversals to converge due to a relatively difficult joint task with no biased initial state. Right: Pre-trained network optimised with combined loss converges at similar rates with different network sizes above 380 units.

## 3.1 EXTRAFIELD PLACE CELL FIRING

First, we investigate the representation learned by the the network during these two stages of training. Pre-training causes the formation of discrete attractors that resemble place cells in the hippocampus. Individual units in the network generally have well isolated place fields, which together cover the whole maze and therefore allow reliable decoding of agent location. In addition to an increase in activity in a particular unit when the agent moves across its respective place field, we also observe substantial extrafield firing of these units. This activity occurs mainly at the primary cue location and at the first choice point after pre-training. After training on the reward task, in addition to the place fields, the network also has units with extrafield activity at the secondary choice points (Fig. 4E).

In the top row of Figure 3(A-D) we show activity in 4 reward trained LSTM units obtained through the collection of unit activity from a full left sided trajectory from the maze start point returning to the start point with cues presented, together with a full right sided trajectory. We show all activity from this activity collection in the top row of Figure 3(A-D) and proceed to outline the maze areas for each unit with activity higher than 30% of the peak activity of that particular unit (mirroring the experimental threshold used by Johnson & Redish (2007)), denoting them as place fields corresponding to these LSTM units. In experiments, rodents seem to pause at high consequence decision points (Johnson & Redish, 2007) with alternating head movement behaviour signifying vicarious trial and error (VTE) (Muenzinger, 1938; Hu & Amsel, 1995). In the activity plots in the bottom row of Figure 3(A-D), we simulate this using our reward trained model by running the agent from the start position at the bottom of the maze stem, then pausing it at the top of the stem, with a left cue presented halfway up. We show activity above 60% of unit peak activity (identified with the previously collected aggregated activity) shown in addition to the previously identified place fields.

The network representation seems to sample both return arms, with surprisingly high extrafield activity in the shown LSTM units when the agent is paused at the maze choice point, a location for which these units do not usually have corresponding activity (Fig. 4A-D). We define nonlocal firing as unit activity above 60% of peak averaged unit activity when running the agent along the central stem (bottom row Fig. 4A-D) to observe only the most poignant extrafield behaviour.

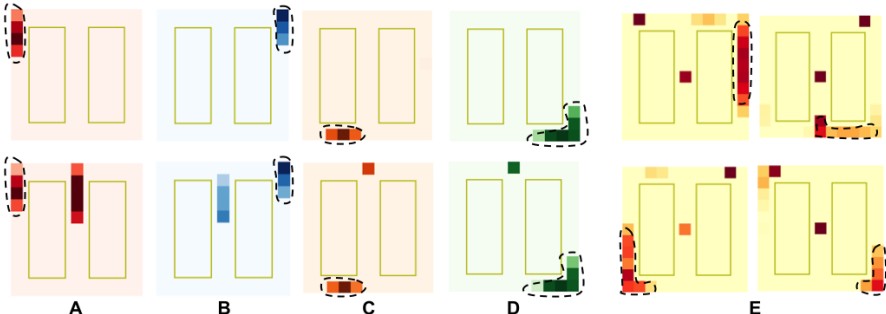

Figure 4: **A-D**) *Top row*: Activity maps showing well isolated place fields of four LSTM units (acting as place cells) indicated in dotted regions after the reward task. Place fields determined by contiguous locality with average activity exceeding 30% peak unit activity during a single left trajectory followed by a right trajectory. *Bottom row*: LSTM unit activity exceeding 60% of previously averaged peak unit activity for the given neuron when agent run from bottom of maze stem to top of stem and given a low frequency (left) cue tone halfway up the stem, then stationary at choice point with LSTM network repeatedly receiving observation from choice point for timesteps thereafter (shown in addition to previously determined unit place fields in dotted regions). **A**, **B**) Strong extrafield firing contiguously from cue to choice point. **C**, **D**) High extrafield firing at choice point while agent is paused at top of stem. **E**) Place fields (determined from average activity on both trajectories) of four LSTM units outlined in dotted areas after reward based task. High levels of consistent extrafield firing at primary and secondary cue points in 56% of LSTM units.

## 3.2 FORWARD MOVING REPRESENTATION

The internal dynamics of the LSTM network has an inherently forward looking representation of the maze once pre-trained in a predictive manner. As depicted in Fig. 5, whilst the agent is stationary, the dynamics of the LSTM network moves forward through the maze, incorporating the trajectory modulation of the cue played halfway up the maze stem. The forward movement of the representation is also notable for having an inconsistent velocity, where the LSTM inferred agent location jumps (Hasselmo, 2009) from the top of the maze to lower down the arm (timestep 16 in Fig. 5).

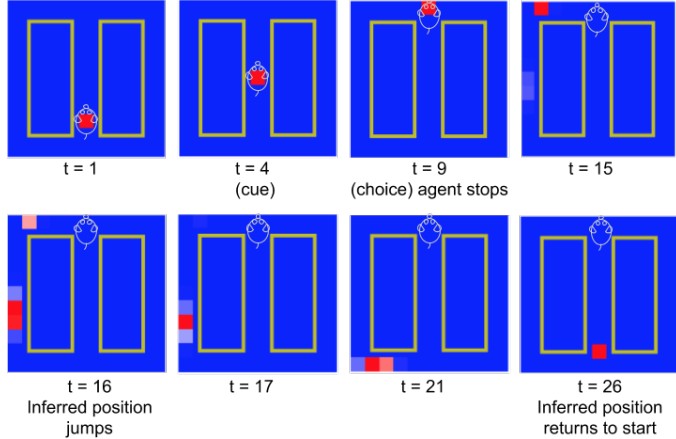

Figure 5: LSTM inferred agent position after pre-training on maze. The agent is run from the start at timestep 1 to timestep 4 where it receives a low frequency cue (indicating a left turn). At timestep 9 the agent is stopped at the top of the maze stem and the LSTM is given the environment observation from this location for the remainder of the shown timesteps. The inferred position then moves left according to the cue with the position seeming to jump abruptly between timesteps 15 and 16. The inferred position then moves back to the starting position at timestep 26. We observe an analogous inferred forward moving representation on the right side of the maze with a high frequency cue.

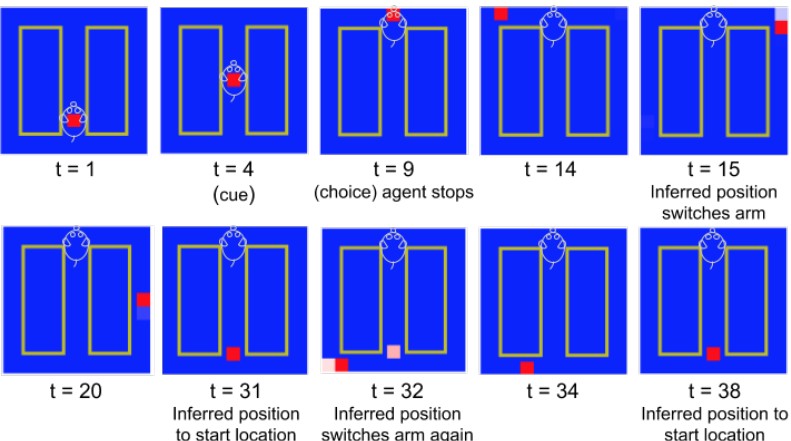

Figure 6: LSTM representation after reward training. As previously, we run the agent from the start position to the top of the stem of the maze at timestep 9 with a low frequency (left) cue tone at timestep 4. Again, the agent is stopped at this position with the LSTM network receiving the environment observation from this position for the remainder of the shown timesteps. As with the network purely trained on the predictive task, the representation moves in the direction corresponding to the frequency of the given cue tone. Then between timesteps 14 and 15, the inferred position jumps from the return arm with active reward sites to the alternate arm, with the inferred position moving from this position to the start location fairly consistently. Then the inferred position jumps again at timestep 32 to the rewarding return arm and moves constantly to the start position.

In stark contrast to the dynamics of the LSTM network after predictive pre-training, following training on the reward task the forward representation of the LSTM is still looking ahead of the agent but is now displaying sweeping behaviour (Fig. 6) which is identified experimentally in rats by Johnson & Redish (2007) when performing cue based tasks. When the agent is stationary at the choice point, we observe the representation moving ahead of the agent - first in the direction corresponding to the cue given at the first cue point and then abruptly down the opposing arm of the maze towards the starting location, thereafter the representation moves down the correct arm (corresponding to the cue) and becomes stationary at the maze start location. This path switching behaviour is reliably observed in networks trained on the combined loss (Eq. 2) with and without pre-training, with differing numbers of units and initial conditions as long as the reward task is solved without backtracking at secondary cue locations. The network lacks a sweeping or forward moving representation when trained on the reward task with Q-learning alone, regardless of pre-training. Thus pre-training does not contribute to sweeping or path switching behaviour.

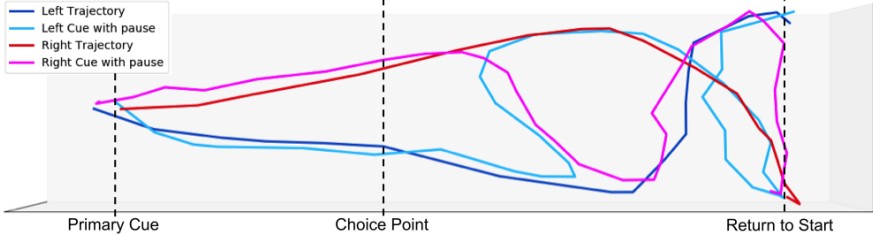

Figure 7: UMAP manifold of LSTM network dynamics of complete left trajectory (dark blue) and complete right trajectory (red) shown along with manifold of dynamics when agent run from start location to choice point with left cue (light blue) and right cue (pink) given at cue point and agent paused in place at the top of the maze stem. A few timesteps after the agent is paused, the dynamics of the left cue paused agent (light blue) switches manifold path abruptly from running alongside the complete left trajectory path (blue) and joins the right trajectory path (red), following this for many timesteps before ultimately resulting at the same manifold end position as the complete left trajectory manifold path (blue). This is analogous for the right cue paths (red and pink).

We further investigate the network representation using Uniform Manifold Approximation and Projection (UMAP) (McInnes et al., 2018). Figure 7 shows generally connected manifolds, with closer inspection revealing the dynamics which leads to the sweeping arm behaviour in Figure 6 when the agent is stationary at the primary choice point. Zeroing visual input while the agent is paused at the choice point gives comparable representation dynamics to that observed in Figures 6 and 7.

Separately, we find that place fields of particular LSTM units drift forwards from their original firing positions after pre-training, towards the reward locations on the return arms throughout reward training, as shown experimentally in CA1 neurons in Lee et al. (2006). We observe this behaviour in 50 out of 380 network units (13%), with final resting locations of place fields at reward locations (seen in Appendix Figure 10). This is possibly explained by the gradient of Q values (prediction of predicted reward) spreading backwards from reward locations (Hasselmo, 2005) and becoming stronger throughout training.

### 3.3 Selectivity of neuronal units

In addition to a forward sweeping representation, this trained network also exhibits neural selectivity that closely matches hippocampal circuits. Griffin et al. (2007) reported that after reward learning, hippocampal neurons were more strongly selective for the encoding or choice phase of a task rather than the direction of the organism's trajectory. We garner the preference of selectivity of each neuronal unit in our network using a discrimination index used by Griffin et al. (2007) for the turn direction selectivity ($DI_{\text{turn}}$) and the phase selectivity ($DI_{\text{phase}}$):

$$DI_{\text{turn}} = \frac{FR_{\text{right}} - FR_{\text{left}}}{FR_{\text{right}} + FR_{\text{left}}} \quad DI_{\text{phase}} = \frac{FR_{\text{cue}} - FR_{\text{choice}}}{FR_{\text{cue}} + FR_{\text{choice}}} \quad (4)$$

where $FR_{\text{right}}$ for a particular LSTM unit is the mean firing rate from the cue point on the central stem to the choice point at the top of the stem on trajectories where the agent turns right at the choice point. Similarly $FR_{\text{left}}$ is the mean stem firing rate when the agent turns left. $FR_{\text{cue}}$ is the firing rate at the cue (encoding) point averaged over both left and right trajectories and similarly $FR_{\text{choice}}$ is the averaged firing rate at the choice (sampling) point.

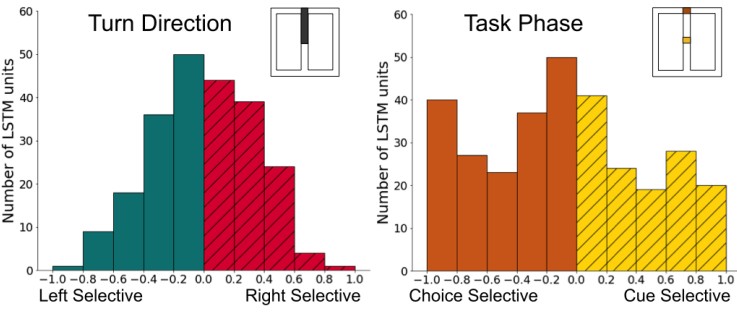

Figure 8: Histograms showing LSTM unit discrimination index for turn direction selectivity ($DI_{\text{turn}}$) vs task phase selectivity ($DI_{\text{phase}}$). A highly negative selectivity index for turn direction indicates a neuronal unit which exhibits high levels of selectivity (uniquely high network activity) for a leftward trajectory and a highly positive selectivity index indicates selectivity for a rightward trajectory. A negative selectivity for task phase indicates a neuron which is highly selective for the choice (retrieval) phase of the goal based task whereas a positive index indicates a neuron which is highly selective for the cue (encoding) phase of the task.

The firing areas used for selectivity measurement are insets in Figure 8. We use the stem above the cue point to assess turn direction selectivity, and the cue/choice points to assess encoding and sampling ($DI_{\text{phase}}$). Figure 8 shows a higher proportion of LSTM units are strongly task selective rather than turn selective, with significantly more units having large absolute $DI_{\text{phase}}$ indices than $DI_{\text{turn}}$ indices.

In addition, the reward trained network is found to have a disproportionately high number of units (163 out of 380 LSTM units) with place fields at the start location of the maze. Moreover, we find

evidence of conditional destination encoding in these units which were heavily differentiated in their firing with respect to particular rewarding locations, as shown experimentally in CA1 hippocampal place cells (Ainge et al., 2007; Wood et al., 2000; Ferbinteanu & Shapiro, 2003). 59.5% of units with a place field at the maze start location fired uniquely at this point for rewarding locations on a particular return arm.

## 4 DISCUSSION

In this work we show that networks trained with a combined predictive and goal-based objective exhibit functional dynamics and selectivity behaviour coinciding with that of hippocampal neurons. We demonstrate that extrafield firing activity of network units emerge when a simulated agent, which is trained on a goal based reward task in a T-shaped maze, pauses at decision points - suggesting intrinsic dynamics are encoding the future trajectory of the agent. This mirrors experimental results in hippocampal place cells in rats (Johnson & Redish, 2007; Frank et al., 2000). At the same time, we find that networks using this combined objective, following pre-training only on a sensory prediction task, can learn the correct goal-directed behaviour much faster than an equivalent network with only a Q learning objective.

Previous work shows that metric neural representations of environments form when an RNN is optimised to predict agent position from agent velocity (Cueva & Wei, 2018; Banino et al., 2018) and non-metric representations form when an RNN is trained to predict future sensory events given direction of movement (Recanatesi et al., 2019). When training our model we do not provide the LSTM network with any explicit information about location or direction, it only receives sensory information. This is similar to the purely contextual input received by the model pre-trained by Xu & Barak (2020) where no velocity input is given, however, the network used by these authors is still trained on position and landmark prediction in a supervised way.

Instead, our training paradigm forces the LSTM to maintain an implicit notion of movement within its internal state in relation to environmental observations. This, in conjunction with the consideration that model-free RL methods such as Q-learning perform poorly on tasks in dynamic environments such as ours (Dolan & Dayan, 2013), and the long term dependency on the delayed cue in perspective of the choice location, makes the task outlined in Figure 2 particularly challenging.

Training on a sensory predictive task causes the formation of a non-metric place cell like representation in the activations of network units, similarly to Recanatesi et al. (2019). These units demonstrate nonlocal extrafield firing (Appendix Figure 9) and after reward training (Figure 4). Johnson & Redish (2007) find that this extrafield firing is particularly striking at consequential decision points where rats usually pause in order to sample previously seen trajectories. We observe that cue or choice point extrafield activity is evident in most LSTM units after training on the reward task. This is likely due to the increased precedence these points have in the agent reaching reward locations. Together the trained LSTM network units form a representation which sweeps along the paths available to the agent, first down the reward path and then the other, as shown in Figure 6 and demonstrated in rats in Johnson & Redish (2007).

Although hippocampal place cells are critical for spatial memory (Nakazawa et al., 2002; Florian & Roullet, 2004; Sandi et al., 2003; Redish & Touretzky, 1998; Miller et al., 2020), it is currently unclear by what mechanism an ensemble of place cells contributes to a representation of goal-directed behaviour (Morris, 1990). Our model and training paradigm is in keeping with the hypothesis that the hippocampus is involved in maintaining a conjunctive representation of cognitive maps and sensory information (Whittington et al., 2019). We show that this paradigm can be extended with predictive learning of Q-values of anticipated future reward, and show that the resulting representation is well suited for learning actions leading from a cue to a reward. Importantly, this representation emerges solely from sampling sensory inputs and predicted rewards, while reinforcement learning itself remains model-free and is initially random. The surprising similarity of the task-dependent activity in our simulations and experimentally recorded neural activity in similar tasks suggests that the model may replicate central aspects of learning and planning in the hippocampus. Our trained model could improve understanding of hippocampal function by testing hypotheses regarding previously unobserved dynamics inexpensively. This could be performed on maze environments such as this work, or more open arena settings once the model is retrained.

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

# A  APPENDIX

## A.1  EXTRAFIELD PLACE CELL FIRING AFTER SENSORY PREDICTION TASK

After pre-training on the sensory prediction task outlined in Figure 1, we observe that when the agent is paused at the top of the stem of the maze the network representation moves far ahead of the agent, caused by extrafield activity of many neurons in the network.

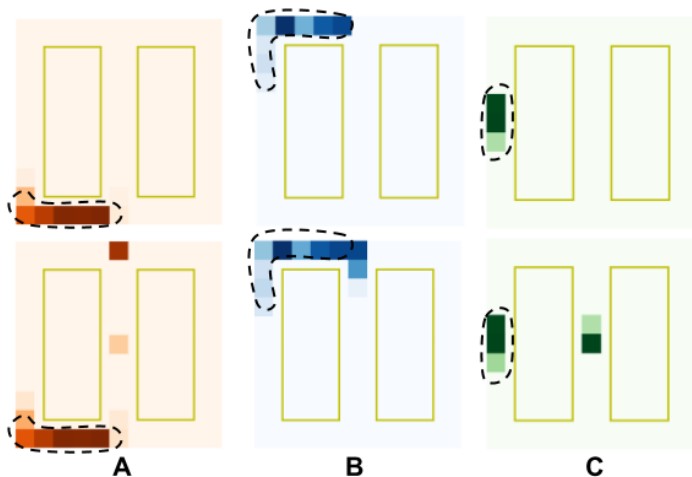

Figure 9: **A**, **B**, **C**) Top row: well isolated place fields of three LSTM units indicated in dotted regions after pre-training. Place fields determined by contiguous locality with average activity exceeding 30% peak field activity during a single left trajectory followed by a right trajectory. Bottom row: agent run from bottom of maze stem to top of stem (and given a low frequency cue tone halfway up the stem) and paused at choice point with LSTM network repeatedly receiving observations from choice point for timesteps thereafter. **A**) Strong extrafield firing at choice point with some activity at the cue point. **B**) Extrafield activity at choice point and at position below. **C**) High extrafield firing at cue point before agent pauses at top of stem.

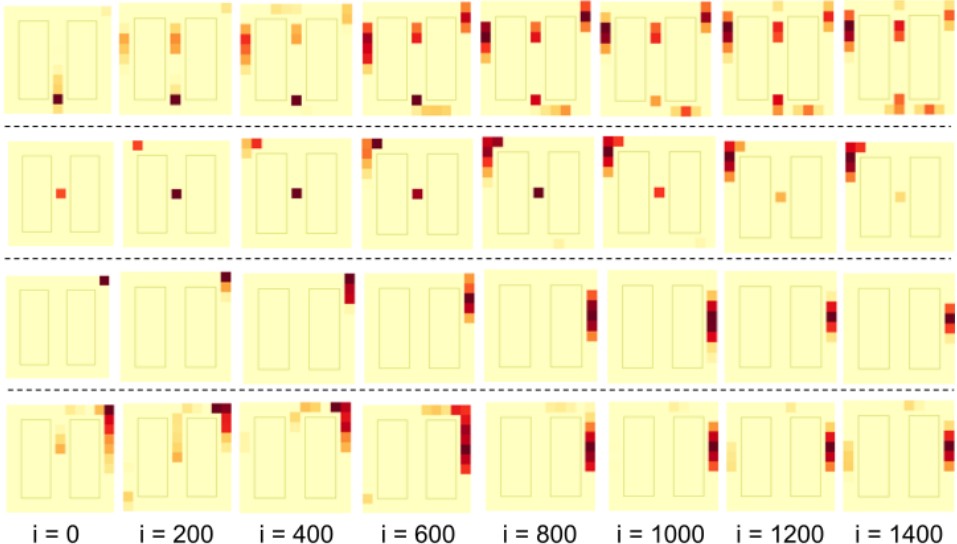

Figure 10: Place fields of four LSTM units, starting from $i = 0$ where the network has been pre-trained on the sensory prediction task, drifting forwards towards reward locations throughout reward training (where $i$ is the number of training iterations). The place fields ultimately rest at maze reward locations at the end of reward training ($i = 1400$).

