# OpenReview forum: "Hippocampal representations emerge when training recurrent neural networks on a memory dependent maze navigation task"
_ICLR.cc/2021/Conference — Reject_

### Official Review · AnonReviewer4 · 2020-10-24

**Rating:** 4
**Confidence:** 4

**Review:**

**Paper summary**

The main goal of this paper is to show that LSTM units in a network trained to solve a T-maze task, show similar activity patterns as neurons in rats solving a similar task. Specifically, the authors make the following claims: (1) an RNN learning the task by  a combination of reinforcement and predictive learning produces internal representations with consistent extrafield firing associated with consequential decision points, (2) the network’s representation, once trained, follows a forward sweeping pattern similar to those found in rats and (3) a higher proportion of units in the trained network show strong selectivity for the choice phase of the task than for spatial topology, as seen in rats.

**Pros**
1. The submission is clear, well-written and the execution is competent.
2. I find the approach well motivated and the problem interesting for the current state of the field.
3. The authors provide a sufficient amount of details so that reproducibility should be possible.

**Cons**

I have concerns about key points of the paper and the interpretation of the results:

1. One of the main results of the paper is the observation that the network produces a forward-moving representation similar to the one observed in rats at the decision point. However, the way the authors simulate this is by freezing the agent at such point and keeping the LSTM running, repeating the same constant observation. The LSTM was trained on trajectories on the maze, so in this trajectory (which was never used during training), the network is completely out of distribution. The activity of any network in this situation is difficult to interpret. Because the network was trained with a predictive loss on two very specific sequences of observations, it seems plausible that it is robust to the change of input statistics and follows the same sequence, maybe with some instabilities. Note that the cue was present, which explains why the correct sequence is followed.
In the network trained also by RL in particular, the authors interpret this as “The agent appears to be sampling the trajectory concerning the alternate return arm of the maze before ultimately settling on the rewarding return arm”. But this is, in my opinion, an over-interpretation, as the agent has no sampling capability in the first place (there is no generative model of observations), nor any particular planning mechanism. Alternatively, the authors may be claiming that this jumping behavior happens only after the RL training and not before, in which case they should emphasize this difference and quantify it explicitly. Although in this case, a simple explanation for this could be that due to the epsilon-greedy, only during the RL training the network is exposed to the wrong cue-arm combination. Therefore, the LSTM would be less able to rely on the cue, which could explain the jump between attractors in the out-of-distribution case. In the discussion section, the authors claim “We demonstrate that extrafield firing activity [..] emerges when a simulated agent [...] pauses at decision points - suggesting intrinsic dynamics are encoding the future planned trajectory of the agent.”. I find this to be an over-claim, as the agent doesn’t pause (it can’t) and doesn’t plan (for any common definition of planning). I would be more convinced if the agent was able to pause (as an additional action) and this behavior was observed in the LSTM activity in this situation, which is closer to the biological case.

2. The pre-train stage is done on trajectories that correspond to the solved task. This means that the LSTM trained by the predictive loss is not exposed to the general structure of the environment but to the specific solution of the task, including the cue-choice association and the exact sequence of observations in each of the two correct trajectories. The authors draw a parallel with the pre-training phase in behavioral experiments (Johnson & Redish, 2007) in which rats usually run each trajectory separately (by having the other one blocked). However, in my opinion, this is problematic for their analysis. First it’s unclear to what extent the network is learning by RL as during the pre-training it has already learnt to predict the observation corresponding to the correct turn (wich corresponds to one of the two actions). Second, the LSTM is exposed only to the correct cue-arm trajectories, which I think is the reason why the forward-looking sweeps only follow these trajectories (see previous point). This is more similar to a demonstration than to pre-training. A more conventional pre-training would leave the agent to explore freely to implicitly learn the structure of the environment (a la Tollman). On the other hand, the argument of following the protocol of the behavioral experiments also doesn’t fully work as the rats are still producing motor outputs and even being rewarded during the pre-training phase (Johnson & Redish, 2007).

3. Finally, a more general concern is the main point of the paper. If I understand correctly, the main claim is the similarity of the observations between the RNN agent and the experimental findings in rats. However, given that there are plenty of arbitrary choices when training an RNN, I believe the results are not particularly explanatory.  I would encourage the authors to formulate better alternative hypothesis and controlled experiments. For example, I would find it interesting to show that the forward-sweeping observations done in rats, which is often interpreted as a signature of planning or prediction of the consequences of future actions, arises simply from a next-step prediction loss in an overtrained rat.

Minor concerns:

- “As such, a network of Gated Recurrent Units [...] or vanilla RNN units was unable to perform well in either the pre-training or joint RL task due to these prevalent long term dependencies.”  How many steps are there between cue and choice, and between choice and reward?

- Related to the previous point: “We attempted to run the reinforcement learning task alone in a maze with no wall colours or environment statistics except the cue. In this scenario the network is not able to learn the task due to a lack of self-localisation.” If I understand correctly, there is a constant number of steps between the cue and the moment where the choice has to be made. I would tend to believe that an LSTM can learn to make a prediction only based on the number of timesteps, regardless of the lack of wall observations (e.g. 2 sequence problem in Hochreiter and Schmidhuber, 1997).

Details:

- It would be good to clarify what exactly is the action set of the agent.
- I would like to know how exactly are activity maps obtained. The authors mention “Place fields determined by contiguous locality with average activity exceeding 30% peak unit activity during a single left trajectory followed by a right trajectory” but I don’t find this particularly clear. I also found it difficult to understand the bottom row of Fig 3.
- Fig 5 should be referred to in the paragraph starting with “In stark contrast to the dynamics of the LSTM network following predictive pre-training...”
- Why is the return to start representation in Fig. 6 different for right and left trajectories?

---

> ### Author Response · Authors · 2020-11-23
> **Response to reviewer 4 (part 1)**
>
> We thank the reviewer for their careful reading of our paper and for their scrupulous evaluation of our results. We have made substantial updates and alterations to the manuscript to address the reviewer’s comments and we respond to feedback below.
>
> Cons:
>
> 1.
>
> * “One of the main results of the paper [...], repeating the same constant observation.”
>
> This is what the rodent is doing experimentally in (Johnson & Redish, 2007), and we replicate this experiment directly, so we feel prudent to follow this as a reasonable simulation for subsequent comparison of hippocampal characteristics. In page 4 of the original manuscript we state that, in experiments, rodents seem to pause at high consequence decision points (Johnson & Redish, 2007) with alternating head movement behaviour signifying vicarious trial and error (VTE) (Muenzinger, 1938; Hu & Amsel, 1995).
>
> * “The LSTM was trained on trajectories on the maze, [...], which explains why the correct sequence is followed.”
>
> As the reviewer states, the activity of the network is difficult to interpret in this situation but we agree this is a plausible interpretation of network dynamics after pre-training and may in fact be what is occurring experimentally.
>
> * “In the network trained also by RL in particular, the authors interpret this as [...], nor any particular planning mechanism.”
>
> We agree that this may be an over-interpretation as the agent has no active sampling capability. Our wording here is ambiguous and we have removed references to agent sampling in the updated manuscript. However, sweeping behaviour (as shown experimentally) by Johnson & Redish is undoubtedly occurring.
>
> * “Alternatively, the authors may be claiming [...] quantify it explicitly.”
>
> The path switching behaviour does only occur after reward training and not before, we have further emphasised this in the paper on page 7. We have updated the paper to indicate that path switching occurs reliably in a very similar way after reward training with differing numbers of LSTM units and initial conditions as long as the reward task is solved without backtracking at secondary cue locations and is trained on the combined loss (Eq. 2, this was Eq. 3 in the original paper). Further analysis of this can be seen in Figure 3 which we introduce in the updated manuscript.
>
> * “Although in this case, [...] out-of-distribution case.”
>
> We would argue that the network is more able to rely on the cue during reward training due to the spatial map of the maze formed during pre-training. Exposure to the wrong-arm combination is not sufficient for the network to exhibit the path switching behaviour shown in Figure 6 for various reasons. Firstly, the network converges to a solution which does not include backtracking at secondary points, therefore the network is choosing actions at the choice points which leads the agent to take a direct path to reward locations. Thus path switching is not part of the network’s inherent behaviour. Secondly, in the out of distribution case when the agent is paused at the primary choice point, the instantaneous representation jump from the rewarding maze arm to the opposing maze arm is not occurring anywhere in training, even at the beginning of epsilon-greedy reward training when actions are chosen completely randomly. Lastly, simply being out of distribution cannot possibly explain the secondary path switch shown at timestep 32 in Figure 6.
>
>
> * “In the discussion section, [...] the biological case.”
>
> We agree that this may be an overclaim and we have removed suggestions of active planning in our updated manuscript, however stark similarity in terms of dynamics to the experimental case is a significant insight we believe. We thank the reviewer for the idea of having agent induced pausing as additional trainable behaviour which would certainly be closer to the biological case. We believe this would be too great a change to the current work as a revision but we will certainly explore this in future work.

---

> > ### Author Response · Authors · 2020-11-23
> > **Response to reviewer 4 (part 2)**
> >
> > 2)
> >
> > * “The pre-train stage is done on trajectories that correspond [...] correct turn (which corresponds to one of the two actions).”
> >
> > The network here must readapt connections in order to predict Q-values rather than RGB values, therefore it must learn by RL. In addition, the actions taken during Q-learning are initially completely random (both at the primary and secondary choice points) and so it must learn by RL entirely to gain perfect performance on the task, as the actions become more on policy over subsequent iterations (epsilon-greedy).
> >
> > Importantly, we show in Figure 3 and on page 4 of our updated manuscript, that Q-learning alone with pre-training learns an altogether different strategy for reaching rewards, contrasting with the case using the combined loss (Eq. 2, this was Eq. 3 in the original paper). Q-learning alone seems to cause the network to completely dissociate from its pre-trained spatial map representation whereas training with the combined loss seems to cause the network to utilise this spatial map effectively to solve the reward task perfectly and in relatively few iterations (Figure 3).
> >
> > * “Second, the LSTM is exposed only [...] of the environment (a la Tollman).”
> >
> > It is not possible to pre-train the agent in the way the reviewer suggests, in a free random exploratory manner. This is due to the structure of the maze and our training paradigm. The complete uncertainty of turn direction when the agent enters or leaves the central maze stem when performing this exploratory random walk method of pre-training would render the network unable to converge with sufficient predictive performance and unable to form a latent spatial map of the maze environment. Our training paradigm consists of the agent only predicting the subsequent visual stimulus based on the previous visual stimulus and does not receive any spatial or velocity information and is not informed of the last action taken either. Therefore pre-training with random movements would be impossible to learn.
> >
> > Importantly, if we pre-train and then train the agent on the reward task using Q-learning alone (and no wall prediction), the representation does not move forward when the agent is paused at the top of the maze. We have updated the manuscript stating this explicitly on page 7. Thus the agent sweep cannot be said to be following these set trajectories. Notably, if we do not pre-train the agent, and train the agent only on the reward task with the joint Q-learning and wall colour prediction loss (Eq. 2), the network representation still recapitulated the same forward sweeping behaviour shown in figure 5 (Figure 6 in the updated manuscript) - thus this behaviour (after reward training) is dissociate from pre-training.
> >
> > * “On the other hand, [...] phase (Johnson & Redish, 2007).”
> >
> > We believe this sort of pre-training (including the Q-learning component) with rewards already being presented would make the task far too easy as the network will have a heavy bias when the reward based training is commenced, therefore we do not see this as conducive to important insights.

---

> > > ### Author Response · Authors · 2020-11-23
> > > **Response to reviewer 4 (part 3)**
> > >
> > > 3)
> > >
> > > * “Finally, a more general [...] overtrained rat.”
> > >
> > > We find that these RNN dynamics are observed under a wide range of initial conditions, training hyperparameters and network sizes, and are recapitulated in a similar fashion in every instance that the network solves the reward task perfectly (without backtracking) and importantly, when the network is trained with the combined loss (Eq. 2) - see explanation on page 7 in the updated manuscript. Thus we believe these results are particularly enlightening with regard to hippocampal dynamics. We have updated the paper showing the variety of network sizes and initial conditions where the reward task is solved (Figure 3 and page 4).
> > >
> > > We find the reviewer’s alternative hypothesis suggestion intriguing, it is indeed possible that sweeping is simply a consequence of recurrent dynamics that resulted from learning the task. We are not aware of published experimental data which could help dissociate a purely dynamical phenomenon from higher level functions such as planning (or if these two are indeed similar in nature). This is an interesting direction where this model could help formulate testable predictions.
> > >
> > >
> > > Minor concerns:
> > >
> > > * “As such, a network of Gated Recurrent [...] between choice and reward?”
> > >
> > > There are 5 steps between the cue and choice points, with 7 steps between the choice point and the first reward location on either return maze arm. We have added these details in the updated manuscript on page 3. The LSTM has a cell state which can be trained to be consistent between timesteps and we believe this is crucial for maintaining the cue modulation in network memory for many timesteps.
> > >
> > > * Related to the previous point: “We attempted to run [...] observations (e.g. 2 sequence problem in Hochreiter and Schmidhuber, 1997).
> > >
> > > We thought this might be the case too (which is why we tested it) but the LSTM is unable to use step information in this way to solve the task. This is due to random actions being taken at all three choice points (primary and two secondary) during epsilon-greedy reward training - this causes the timestep number between choice points and prior taken actions to be inconsistent in various maze traversals during reward training.
> > >
> > > Details:
> > >
> > > * It would be good to clarify what exactly is the action set of the agent.
> > >
> > > The LSTM predicts Q-values relating to four agent actions, movement either up, down, left or right (cardinal directions). It predicts all four Q-values regardless of whether they can be used at a given position or not - the allowed action with the highest Q-value is chosen at each step when the action is not chosen randomly. We already mention that the network predicts state-action values associated with the four cardinal directions in relation to the agent’s current position and direction. We have updated the manuscript mentioning the network chooses optimal actions of those available.
> > >
> > > * I would like to know how exactly are activity maps obtained. The authors mention “Place fields determined by contiguous locality with average activity exceeding 30% peak unit activity during a single left trajectory followed by a right trajectory” but I don’t find this particularly clear. I also found it difficult to understand the bottom row of Fig 3.
> > >
> > > Activity maps are obtained by running the agent and collecting activity from the start point of the maze through a left sided trajectory back to the start point, followed by a right sided trajectory (with cues presented) back to the start point. This is what we show in the top row of figure 3 (Figure 4 in updated manuscript). We then denote place fields as areas of firing where the aggregated activity exceeds 30% of the peak activity level for each particular LSTM unit - the dotted regions in the top and bottom rows of figure 3 (Figure 4 in updated manuscript).
> > >
> > > For the activity maps in the bottom row of figure 3 (Figure 4 in updated manuscript), we run the agent from the start point of the maze to the top of the central maze stem (presenting a left side cue at the cue point) and pause the agent here feeding the LSTM the same observation input for many timesteps. Here we show activity exceeding 60% of the peak activity level for each LSTM unit in order to emphasise that this extrafield firing phenomenon occurs at a significant level. This is shown in addition to the previously identified place fields (as in the top row).
> > >
> > > We have clarified this in depth on page 5 of the updated manuscript.
> > >
> > > * Fig 5 should be referred to in the paragraph starting with “In stark contrast to the dynamics of the LSTM network following predictive pre-training...”
> > >
> > > We thank the reviewer for this suggestion and we have added this reference on page 7 of the updated manuscript.
> > >
> > >
> > > * Why is the return to start representation in Fig. 6 different for right and left trajectories?
> > >
> > > We believe this is due to left and right trajectories having somewhat dissociated within the LSTM representation after reward training.

---

### Official Review · AnonReviewer3 · 2020-10-27
**Unifies prior approaches though not many new insights**

**Rating:** 7
**Confidence:** 4

**Review:**

Motivated by biological considerations, this paper shows that recurrent networks trained with a predictive and goal-based objective on a maze finding task, qualitatively recapitulate experimental findings in hippocampal recordings in rodents trained on the same task. In particular, these LSTM networks demonstrate both metric representations of their environment and nonlocal extrafield firing at decision points along the maze (anticipating the future trajectory of the agent).

Strengths:
+ I like that the authors take a normative approach that exhibits both metric and non-metric place cell representations of the environment, unifying prior findings in one model.
+ I appreciate that no velocity input is given to their model, in contrast to prior approaches.
+ I also liked the qualitative comparisons to hippocampal recordings from rodents trained on the same task (especially Figures 6 and 7).

Weaknesses:
- The primary conclusion, namely, training an RNN on a maze-like environment gives you place cells, is really not all that new, especially considering that the network is still supervised to predict position and landmarks.
- Given that lack of novelty in the modeling conclusion, it would have therefore been nice to have seen more quantitative comparisons to hippocampal recordings in rodents. Does their approach explain more variance in these neurons than prior approaches? Otherwise, it seems that they simply recapitulate prior qualitative comparisons.

Minor comments: “Recurrence based” should be changed everywhere to “recurrent” (e.g. on pg. 1), and “Neuroscience” is not capitalized. The motivation to use double Q learning should be expanded on in pg. 3 prior to equation 3.

Question: The authors mention that Q-learning performs poorly on tasks in dynamic environments – however, I do not see any evidence of this in the paper, it would be imperative to show this explicitly for the environments they consider. Suppose this is in fact the case, could you clarify what makes your approach more successful at this task than others? Is it because of the pretraining to predict the subsequent observation of wall colors from the current wall color observations?

As it stands, I think the ideas of this paper are interesting and think it unifies prior approaches, but I do not think the conclusions from the modeling add all that much novel insight from prior approaches. Therefore, I recommend a weak accept.

---

> ### Author Response · Authors · 2020-11-23
> **Response to reviewer 3**
>
> We thank the reviewer for their feedback and overall positive evaluation of our work. However, we believe the reviewer may be unfamiliar with the previous work in this area. We address the points made below and in our updated manuscript.
>
> Weaknesses:
>
> 1. The primary conclusion, namely, training an RNN on a maze-like environment gives you place cells, is really not all that new, especially considering that the network is still supervised to predict position and landmarks.
>
> We think the reviewer is incorrect here, to the best of our knowledge training an RNN on an environment with emerging place cells has only been shown once before (Recanatesi et al., bioRxiv 2019) and for the first time in a maze-like environment in this work. Our main result is not purely the emergence of place cells, but the combination of predictive and reinforcement learning to solve a navigation task which results in network units displaying hippocampal neuron characteristics.
>
> It is important to point out that the network in our model is not supervised to predict position or landmarks at all, it is only instructed to predict subsequent visual stimuli in pre-training and extended to predict global reward in Q-learning. We think the reviewer may have been misled by this line in our discussion section: “This is similar to the purely contextual input received by the model pre-trained by Xu & Barak (2020) where no velocity input is given, however, the network here is still trained on position and landmark prediction in a supervised way.”
>
> To clarify, here we are referring to the work by Xu & Barak, not our work. We have updated our paper to make this clearer.
>
> Our hypothesis was that predictive learning is well suited to be combined with reinforcement learning (we add a prediction of the Q-value for reward training). We show that the resulting representation not only allows efficient learning but also yields cells with properties observed in the hippocampus. It is important to stress that previous work where training was based on trajectories of velocity and position did not report such behaviour and only demonstrate the formation of activations with the form of entorhinal cortex grid cells which are used in navigation differently to place cells. These authors also did not analyse resulting dynamics.
>
> 2. Given that lack of novelty in the modeling conclusion, it would have therefore been nice to have seen more quantitative comparisons to hippocampal recordings in rodents. Does their approach explain more variance in these neurons than prior approaches? Otherwise, it seems that they simply recapitulate prior qualitative comparisons.
>
> We agree with the reviewer that direct quantitative comparisons would be interesting yet they may be misguided given the significant differences between neural and neuronal networks. We believe in this context our main novel contribution is the demonstration that the combination of predictive and reinforcement learning results in dynamics also observed in the hippocampus, specifically we show extrafield firing of network units at locations outside of their apparent place fields (Johnson & Redish, 2007), non-local forward sweeping representation of the network (Johnson & Redish, 2007), place fields drifting towards reward locations throughout training (Lee et al., 2006), a high proportion of units with place fields at the maze start location encode reward locations (Ainge et al., 2007) and that a higher proportion of units encode task phase than turn direction (Griffin et al., 2007). These specific comparisons between RNNs and hippocampal neurons have not been demonstrated before as far as we are aware.

---

> > ### Author Response · Authors · 2020-11-23
> > **Response to reviewer 3 (continued)**
> >
> > Minor comments:
> >
> > * “Recurrence based” should be changed everywhere to “recurrent” (e.g. on pg. 1), and “Neuroscience” is not capitalized. The motivation to use double Q learning should be expanded on in pg. 3 prior to equation 3.
> >
> > We thank the reviewer for pointing out these errors and these have been corrected in the updated manuscript. We have also motivated the use of double Q-learning in the updated paper on page 4.
> >
> > * Question: The authors mention that Q-learning performs poorly on tasks in dynamic environments – however, I do not see any evidence of this in the paper, it would be imperative to show this explicitly for the environments they consider. Suppose this is in fact the case, could you clarify what makes your approach more successful at this task than others? Is it because of the pretraining to predict the subsequent observation of wall colors from the current wall color observations?
> >
> > We thank the reviewer for highlighting this (as also done by reviewer 2).
> > In the paper we state that Q-learning alone cannot solve the task after the network has been pre-trained on the predictive task. This is the case when using the rate of epsilon decay (in epsilon-greedy RL training) we use for the joint Q-learning and wall colour prediction training (Eq. 2, this was Eq. 3 in the original paper). We have run more rigorous analyses of running Q-learning alone with the same network without pre-training, Q-learning alone with pre-training and the combined Q-learning and colour prediction loss with (method presented in paper) and without pre-training. These analyses can be seen on page 4 and in Figure 3 of our updated manuscript. In essence, training with a combined loss (Eq. 2) after predictive pre-training converges with far fewer training iterations than any of the other cases. This shows that predictive learning builds efficient representations that contain relevant task variables, without including these explicitly in the optimisation procedure.
> >
> > * As it stands, I think the ideas of this paper are interesting and think it unifies prior approaches, but I do not think the conclusions from the modeling add all that much novel insight from prior approaches. Therefore, I recommend a weak accept.
> >
> > We thank the reviewer for their recommendation but would like to stress again that our model is markedly different from previous approaches, and reproduces key aspects of hippocampal dynamics which have not yet been shown in a neural network.

---

### Official Review · AnonReviewer2 · 2020-10-28
**Interesting findings, bad explanation.**

**Rating:** 5
**Confidence:** 5

**Review:**

Summary:

The authors trained a recurrent network to perform a sensory prediction task and this gave rise to units that resembled hippocampal place fields. Then they augmented the network with a Q-learning objective and shown that the activity in the network sweep forward in space if the agent is fixed at a decision point.

##########################################################################

Reasons for score:

Overall, I think the paper should be rejected. The work is interesting, but the clarity of the paper is not at the level of the findings. I think the authors should enhance exposition, and strengthen some analysis. The findings are really interesting ,but at the current stage I don’t think the paper is ready to be published. However, I’m happy to revise my score if authors addresses my comments.

##########################################################################

Pros:

1. Combination of unsupervised learning and RL to show that units in a recurrent network can be used to understand spatial and non-spatial firing patterns in the hippocampus

2. Figure 4 and 5 are convincing.


Cons:

1. While explaining the task the authors claim that the colours are chosen at random, however it is not clear whether these stay fixed across episodes or they changed. Intuitively it looks like, once the colour are generated, then they stay fixed, but in this case it is difficult to understand why a simple Q-leaning objective won’t be able to solve this task (as claimed by the author). The only reason I can think about is if the steps between the cue and the reward are longer that the ones allowed by the discount factor chosen (0.8). However, in the paper there are no details about the number of steps or how the discount affect the results. I think this is a serious issue.

2. The authors claim that they first pre-train on the predictive task, but then in the loss of eq. 3 they report a combined loss. Does it mean that the loss_{rgb} is also fine-tuned while training with the Q-learning objective? This point need clarification.

3. The paper doesn’t report any details about the learning rates or the sizes of the linear layers used in eq. 1 and 2. This way is impossible to replicate this results. This is a serious issue.

4. How the threshold for place cells are defined? Why 30%? How many units show firing above that threshold? This needs further analysis to support the decision, which otherwise seem very arbitrary.

5. Are figure 4 and 5 just cherry picked run or are these averaged across several testing runs? Also in the captions of these figures the authors are using the singular “observation ” in figure 4 and the plural “observations” in figure 5. Does it means that the analysis have been performed differently? Or is it just a mistake?. This is an important point as I think the results will be more powerful with the same image fed as input, or even better with no image, just with 0ed input to simulate pondering.

Minors:

1. 380 units seems quite an unconventional number of units, why not 256 or 512? Can you please explain. Also is this number affecting the representations? Have you done a sweep and settle on this number because it support your findings better? If so, it would be important to mention it.

2. It would have been nice to have analysis to support the following sentence on page 4: Generally, when the network loses its ability to self-localise the agent, state-action values are no longer reliable indicators of future reward potential as the current environmental state is not clearly discernible. Otherwise please correct it.

---

> ### Author Response · Authors · 2020-11-23
> **Response to reviewer 2**
>
> We thank the reviewer for their feedback and for being transparent with regards to their evaluation. We have clarified the points made and have made substantial updates to the manuscript to address the reviewer’s comments.
>
> Cons:
> 1. Wall colours fixed or variable, Q-learning objective solving this task, steps between the cue and the reward and discount factor chosen (0.8):
>
> We very much agree with the reviewer that this is ambiguous in our current explanation and we have updated the paper (page 2), explicitly stating that the wall colours remain fixed throughout training. In the paper we state that Q-learning alone cannot solve the task after the network has been pre-trained on the predictive task. This is the case when using the rate of epsilon decay (in epsilon-greedy RL training) we use for the joint Q-learning and colour prediction training (Eq. 2 - Eq. 3 in original paper) described in the paper.
>
> We have run more rigorous analyses of running Q-learning alone with the same network without pre-training, Q-learning alone with pre-training and the joint Q-learning and colour prediction loss (Eq. 2) both with pre-training (shown in the paper) and without pre-training. We have updated the paper (Figure 3, page 5) to summarise these results. Taken together, training with a joint loss after pre-training converges with far fewer training iterations than the other cases.
>
> We have previously run analyses with differing discount factor values and find that a value higher than 0.8 regularly causes the model to converge without taking the most direct route to reward locations (i.e converging on solutions with backtracking at secondary choice points when discount factor is higher than 0.8).
> The number of steps in a single training episode is 30 (if the agent takes the most direct path without backtracking). There are 5 steps from the cue to the choice point with 7 steps from the choice point to the first reward site. We have updated the paper (page 4 and page 3) explaining the discount factor used and the number of steps inherent in the task.
>
> 2. Loss_rgb fine-tuned while training with the Q-learning objective:
>
> We agree this point is ambiguous and we have clarified this in the paper (page 4). The predictive pre-training task where loss_rgb is optimised converges completely. It is not fine-tuned while training the Q-learning objective to navigate to reward locations, but is required so the non-metric representation of space is maintained during learning. When loss_rgb is not included during Q-learning, the place fields are lost, and the training converges much slower to a different solution without an explicit representation of space.
>
> 3. Learning rates and sizes of linear layers used in eq. 1 and 2:
>
> We have updated the paper to include the following details on pages 2, 3 and 4 of the manuscript. The linear layers in eq. 1 and eq. 2 (Eq. 3 in the updated manuscript) are simply single layered readout layers for the LSTM. To clarify, when we have a 380 unit LSTM, the shapes of these readout layers are 380 x 12 (when predicting four RGB wall colours) and 380 x 4 (when predicting Q-values) respectively. We use a learning rate of 0.0005 with an Adam optimiser for reward training, as we find this learning rate gives good convergence (without backtracking at secondary points) with a greater range of training hyperparameters and initial conditions than with a learning rate of 0.001. We use a learning rate of 0.001 for pre-training.

---

> > ### Author Response · Authors · 2020-11-23
> > **Response to reviewer 2 (continued)**
> >
> > Cons... :
> >
> > 4. Threshold for place cells are defined? Why 30%? How many units show firing above that threshold?
> >
> > The reviewer is correct in pointing out that this threshold is arbitrary but is not clear on how this threshold is applied - we have updated the paper (page 5) to clarify this.
> > A threshold of 30% is used to mirror the threshold used experimentally by Johnson and Redish at a per place cell level. It is a threshold placed at a per LSTM unit level in our work, so we denote a unit’s place field as the unit activity above 30% of the peak activity of each particular unit - these are shown in dotted regions in figure 3 (now figure 4 in the updated manuscript). Every LSTM unit has a different firing threshold.
> >
> > We obtain the activity maps in the top row of figure 4 through the collection of unit activity from a full left sided trajectory from the start point returning to the start point with cues presented together with a full right sided trajectory. We show all activity from this activity collection in the top row of figure 4 and proceed to outline the areas in each unit with activity higher than 30% of the peak activity of that particular unit.
> >
> > In the activity plots in the bottom row of figure 4, we only show activity above 60% of unit peak activity (identified with the previously collected activity) when the agent is run from the start point to the top of the central maze stem and paused, with a left cue presented halfway up the maze stem, shown in addition to the previously identified place fields.
> >
> > 5. Cherry picked runs or are these averaged across several testing runs? Singular or plural observations? 0ed input to simulate pondering?
> >
> > The reviewer notes an important point on the reliability of the forward sweeping and path switching phenomena shown in figures 4 and 5 (now figures 5 and 6 in the updated manuscript). Reviewer 1 has also requested clarification on this point. Forward moving behaviour of the network representation (as seen in Figure 5) occurs after pre-training consistently with regards to initial conditions and the number of units in the network. Path switching behaviour exhibited in Figure 6 arises in every instance where the network converges and solves the reward task with a combined loss (Q-learning and predictive), and without backtracking at secondary choice points. This occurs robustly with different network sizes above 380 units and initial conditions as long as the target Q network is updated every 15 or so iterations and the discount factor is low enough so the network does not converge on solutions which include backtracking.
> >
> > We thank the reviewer for pointing out this erroneous caption and have corrected this in the paper. Indeed the network is fed the same visual input while the agent is paused and analysis is performed identically in both figures. We appreciate the reviewer’s suggestion to completely zero this input to simulate pondering - we find intriguingly comparable results in this case and have updated the paper (page 8) to include this outcome.
> >
> > Minors:
> >
> > 1. Why 380 LSTM units instead of 256 or 512? Is this number affecting the representations?
> >
> > We found that the network converges and solves the reward task without backtracking at secondary choice points with 380 or more units, and we used this size (380) to show the efficacy of the model with as few LSTM units as possible. We show our analysis on convergence in Figure 3 in our updated manuscript. The representations and behaviours emerging as a result of training (figures 3, 4 and 5) (now figures 4, 5 and 6) are highly comparable with different network sizes above 380 as long as convergence without backtracking at secondary points is achieved (which generally does not occur with a network smaller than 380 units). We clarify these points in our updated manuscript on pages 4, 5 and 7.
> >
> > 2. It would have been nice to have analysis to support the following sentence on page 4: “Generally, when the network loses its ability to self-localise the agent, state-action values are no longer reliable indicators of future reward potential as the current environmental state is not clearly discernible”.
> >
> > We agree with the reviewer that this sentence is ambiguous and have decided to remove it in our updated manuscript. We instead opt for more analysis on different training regimes and how these are affected by pre-training - we have added this on page 4 and Figure 3.

---

### Official Review · AnonReviewer1 · 2020-10-28
**A compelling model of some hallmark properties of the hippocampus**

**Rating:** 7
**Confidence:** 4

**Review:**

In this paper, the authors train a recurrent neural network on a navigation task, and observe the emergence of several phenomena reminiscent of the hippocampus: appearance of place cells with a secondary receptive field at task-relevant locations; anticipation of possible future paths in the activity of the model, with alternation in time between possible future paths; a high proportion of neurons tuned to task variables rather than animal trajectory.

Strong points:
- these findings are compelling, they account for some hallmark properties of the activity of hippocampus, and they could lead to a better understanding of the role and function of the hippocampus.
- the experiments are rigorous and convincing.

Weak points:
- some technical aspects of the paper could be clarified (see below)
- it is unclear how this model improved our understanding of the hippocampus function, and whether the model makes any testable predictions about the hippocampus.

I recommend to accept this paper because of its strengths listed above.

Clarification questions:
1) I did not understand the role of secondary cue point. Why were these required in addition to the primary cue point and choice point?
2) vocabulary: What are "metric" and "non-metric" representations?
3) How reliable is the alternative path visiting phenomenon? Can this phenomenon be observed reliably in networks trained from different initial conditions?
4) I did not understand the consequences/take-homes of the second and third paragraph discussions.

Additional feedback:
1) It would be interesting to see a discussion on what we learned about the function of the hippocampus from this model, and/or what predictions this model makes about neural activity in hippocampus.
2) Are there any oscillations and phase-precessions phenomena in the model? If not, it would be interesting to discuss why these oscillations might be present in the brain but not in the model.
3) Could this network be used to simulate difficult experiments, e.g. understand how future paths exploration works in an open-field setting?
4) The first sentence of the abstract is difficult to understand. In general, shorter sentences could improve clarity.

---

> ### Author Response · Authors · 2020-11-23
> **Response to reviewer 1**
>
> We thank the reviewer for their careful reading and positive evaluation of our paper, in addition to the constructive feedback. We have clarified the points raised and updated the manuscript accordingly.
>
> Weak point: Using our model to improve understanding of the hippocampus and make testable predictions:
>
> We fully agree that this would be the ultimate goal of our model, to improve understanding of the brain with a relatively small RNN model which can be used to test hypotheses regarding hippocampal dynamics. In this work we aim to show that the LSTM dynamics resulting from training the network using the combination of predictive and reinforcement learning on the maze reward task, mirrors hippocampal neuronal dynamics found experimentally, and as such provides evidence that the underlying learning rules may be similar in the hippocampus. Specifically, we show that non-metric attractors form in the activation space of our network units in the way of place cells, we show extrafield firing of these units at location outside of their apparent place fields, non-local forward sweeping representation of the network, place fields drifting towards reward locations throughout training, a high proportion of units with place fields at the maze start location encode reward locations and that a higher proportion of units encode task phase than turn direction. As far as we know, this is the first model to replicate all these behaviours, and as such provides evidence that the underlying learning rules may be similar in the hippocampus. We therefore expect that our model can be used to generate new predictions in other tasks, and are currently working on this question. We have added a section on how our model could improve hippocampus understanding in the discussion section of the paper.
>
>
> Clarification Questions:
>
> 1. Role of the secondary cue:
>
> The reviewer makes an important point on the necessity of the secondary cue points in this task. Our overall aim is to compare our trained model’s dynamics with that of hippocampal neurons as captured using experimental data. To this end we aim to mirror these experimental set ups as closely as possible - in this case we mirror the maze set up of Jonson and Redish, 2007 in order to optimally compare our resulting dynamics to that of hippocampal neurons. In terms of the task, including secondary cue/choice points gives the agent the opportunity to backtrack on its decision made at the primary choice point in light of further environmental observation (the presentation or lack thereof of the secondary cue). We believe that experimentally, the presence of secondary cue points gives rise to some of the extrafield firing observed by Johnson and Redish as the rodent reevaluates its prior primary choice at a secondary cue point, resulting in the firing of place cells with place fields on the opposing side of the maze. We have updated the paper (page 3) justifying the inclusion of the secondary cue.
>
> 2. Metric and non-metric representations:
>
> Metric representations relate to a Euclidean spatial map of an environment and are biologically akin to grid cells in the entorhinal cortex, whereas non-metric representations relate to associative landmark maps of the environment and are comparable to place cells in the hippocampus. We feel this is adequately explained in the introduction.

---

> > ### Author Response · Authors · 2020-11-23
> > **Response to reviewer 1 (continued)**
> >
> > Clarification Questions... :
> >
> > 3. Reliability of path switching phenomenon:
> >
> > This is an important point which we should have touched on in the paper. This phenomenon is reliably observed from different initial conditions and network sizes, and is apparent when reward training of the network converges using a combined loss (Q-learning and wall prediction together - Eq. 2) and solves the task perfectly (i.e no backtracking at secondary cue/choice points). The network generally converges in this manner as long as the target Q predictor network is updated around every 15 iterations and the network contains more than 380 units (see Figure 3). We have updated the paper (page 7) reflecting on the reliability and conditions of the path visiting phenomenon and look at convergence with differing network sizes (page 4).
> >
> > 4. Consequences of second and third paragraphs in discussion:
> >
> > In these paragraphs we aim to emphasise why our training paradigm differs from previous work and why this difference may lead to the resulting dynamics shown in the paper. Previous work (those with emerging grid cells as a result of training) consists of recurrent networks which are given positional and velocity information and this culminates in responses of the units similar to grid cells. This sort of training paradigm allows for a straightforward emergence of metric representations such as grid cells, however our model only receives visual stimulus and only learns to predict the subsequent visual stimulus, the inputs do not contain any spatial information. We then extend this to a model which predicts subsequent visual stimulus and Q-values from the same input, and thus propose that the hippocampus generally functions as a predictive network. We have extended the text to further clarify these points.
> >
> > Additional Feedback:
> >
> > 1. Discussion on what we learned about the function of the hippocampus from this model, and model predictions about neural activity in hippocampus.
> >
> > Our main focus in this paper is on the learning architecture we propose, and how this not only efficiently learns to solve a RL task, but also reproduces several properties of hippocampal place cells. For further clarification, please also see our replies above. We agree it would be interesting to use this model to generate testable predictions, and have added some thoughts on this in the discussion.
> >
> > 2. Oscillations and phase-precessions phenomena in the model:
> >
> > The LSTM network we use and artificial neural networks in general are not based on spiking communication, therefore we could not observe this phenomena in our model, however we will try with spiking networks in future work. As it stands, our model is not capable of producing oscillations and the chunking of activity into theta cycles, therefore it is difficult to address the question whether it exhibits a phenomenon similar to phase precession.
> >
> > 3. Network used to simulate difficult experiments and understand how future paths exploration works in an open-field setting:
> >
> > The reviewer touches on an important extension to this work which we are already looking into. We use this model to train an agent on an open arena to navigate towards reward locations, pre-training the agent to perform random walks in the open arena and predicting subsequent visual stimuli as in this work beforehand. We observe the formation of well isolated singular place fields in the majority of network units and we will next explore the dynamics of the trained network to attempt to understand path exploration.
> >
> > 4. First sentence of the abstract is difficult to understand:
> >
> > We agree with the reviewer and we have divided this sentence into 2 parts in the updated manuscript. We believe this summarises the goals of the paper well.

---

### Author Response · Authors · 2020-11-23
**Statement to all reviewers: Paper revision uploaded with new analyses and improved explanations**

We thank all four reviewers for their attentive reading of our paper and for their positive evaluations and constructive feedback. We trust that we have addressed all comments raised by reviewers in our individual replies below, and we have run the following analyses to further clarify concerns:

* We have conclusively compared the training time of four LSTM training paradigms: the combined loss shown in the paper (Eq. 2), with and without pre-training and Q-learning loss alone with and without pre-training (Figure 3 and page 4).
* We have tested the reliability and required conditions of the path switching phenomenon shown in Figures 6 and 7.
* We have compared the training time and convergence rate for the pre-trained combined loss model with different network sizes (Figure 3) and also tested the reliability of path switching in these resulting representations.


In addition we have made the following major changes to the paper:
* Justification for the inclusion of secondary cue points in reward training (page 3).
*Explanation for the use of double Q-learning and justification for use of a discount factor of 0.8 (page 4).
*Explanation for the inclusion of the predictive component in the combined loss function (page 4).
*Vastly improved explanation of Figure 4 extrafield activity map generation (page 5).
*Explanation of required minimum training conditions for emerging path switching behaviour of network representation (page 7).
*Analysing the effect of zeroing visual input while the agent is paused at the choice point (page 8).
*Discussion on how our model could be used to improve the understanding of the hippocampus (page 9).

We have uploaded an updated manuscript with these analyses and alterations, and with all reviewer comments answered. The newly performed analyses do not change the conclusions of the paper, but we think both strengthen and extend them. We’d also like to reiterate the primary outcomes and novelties of the paper:

* We introduce a novel training paradigm combining predictive and reinforcement learning which converges in far fewer iterations on a reward task after predictive pre-training vs. reinforcement learning alone.
* This training paradigm replicates key observations from hippocampal place cells:
    1. Non-metric attractors form in the activation space of our network units in the way of place cells, uniformly covering the maze environment.
    2. Extrafield firing of these units at locations outside of their apparent place fields.
    3. Non-local forward sweeping representation of the network.
    4. Place fields drifting towards reward locations throughout reward training.
    5. A high proportion of network units with place fields at the maze start location encode reward locations.
    6. A higher proportion of network units encode task phase than turn direction.

As far as we know, this is the first model to replicate these behaviours in a neural network.

---

### Decision · Program_Chairs · 2021-01-07
**Final Decision**

**Decision:**

Reject

**Comment:**

This paper analyses a recurrent neural network model trained to perform a simple maze task, and reports that the network exhibits multiple hallmarks of neural selectivity reported in neurophysiological recordings from the hippocampus— in particular, they find place cells which also are tuned to task-relevant locations, cells which anticipate possible future paths, and a high proportion of neurons tuned to task variables.

The reviewers appreciated the interesting empirical analysis, and the demonstration that multiple such features could arise in the same neural network— to the best of my knowledge, this had not been demonstrated explicitly before. However, there were also multiple concerns, which lead to this paper beeing discussed extensively and controversially. In particular, it is not clear which features arise from which learning objective, for example, for place cells to arise, do we  just need sensory prediction, or do we need q-learning? In addition, there were some points in which the tightness of the analogy between model and biology is questionable— in particular, this refers to the comprising between hippocampal recordings and the evaluation of the network.  Finally, it is also clear that  some of these observations reported in the paper are, indeed, empirical observations rather than explanations. Because of these shortcomings, there was no consensus and strong support from the reviewers for acceptance of the paper.

After extensive discussion between both the reviewers, the AC and the program chair, the final decision was to not accept the paper. We do hope that the reviews will help you in improving the study and its presentation. It clearly has potential to be a valuable contribution to the literature.